



# Variability of stratospheric aerosol size distribution parameters between 2002 and 2005 from measurements with SAGE III/M3M

Felix Wrana[1], Terry Deshler[2], Christian Löns[1], Larry W. Thomason[3], and Christian von Savigny[1]

[1]Institute of Physics, University of Greifswald, Felix-Hausdorff-Str. 6, 17489 Greifswald, Germany
[2]Department of Atmospheric Science, University of Wyoming, Laramie, Wyoming, USA
[3](retired) NASA Langley Research Center, Hampton, Virginia, USA

**Correspondence:** Felix Wrana
(felix.wrana@uni-greifswald.de)

**Abstract.** Stratospheric aerosol size distribution parameters are derived from the solar occultation data of the SAGE III/M3M instrument and their evolution between 2002 and 2005 is shown. The broad wavelength spectrum of the measurements allows for the retrieval of all parameters controlling the assumed monomodal lognormal size distribution. Besides including periods with very close to background conditions, there were three smaller tropical eruptions during the SAGE III/M3M mission.

After the Ruang, Reventador and Manam eruptions a reduction in average aerosol size and an increase in number density was observed. Apart from the likely effect of the eruptions on the particle size distribution (PSD), an influence of seasonal polar winter condensation events including meteoric smoke particles on the retrieved aerosol size is possible, especially due to the longlasting low stratospheric temperatures during the northern winters of 2002/2003 and 2004/2005. During the same winters, polar stratospheric clouds (PSCs) were likely observed by the instrument. A comparison of the stratospheric aerosol size

retrieval data set with balloon-borne in situ measurements in Kiruna, Sweden, shows generally good agreement, but there are systematic differences between in situ and satellite retrievals below roughly 15 km altitude. Finally, the effect of the necessary assumption of a PSD shape on the aerosol size retrieval with remote sensing instruments is shown and discussed.

## 1 Introduction

In the lower stratosphere there exists a global and permanent layer of aerosol particles roughly between 15 and 30 km altitude.

It was first measured and described by Junge et al. (1961) through balloon-borne measurements and is therefore called the Junge layer. The aerosols were subsequently found to be liquid droplets mostly composed of sulfuric acid and water, sometimes containing traces of other components such as meteoric and carbonaceous material (Rosen, 1971; Murphy et al., 2013; Kremser et al., 2016).

Stratospheric aerosols have effects on the chemistry of the stratosphere by providing surfaces for heterogeneous reactions,

which is important in, e.g., ozone chemistry (Hofmann and Solomon, 1989; Gleason et al., 1993; Deshler, 2008). Furthermore, they affect Earth's radiation budget by scattering shortwave and absorbing and emitting long-wave radiation. This way stratospheric aerosols warm the stratosphere and cool the Earth's surface (Lacis et al., 1992).



In unperturbed background conditions the Junge layer is maintained through a more or less continuous flux of sulfur contain-
ing precursor gases from the troposphere, such as $SO_2$ and most importantly OCS, which are eventually oxidized to sulfuric
acid (Kremser et al., 2016), forming small droplets through co-condensation with water vapor (Curtius, 2006). However, the
variability of the stratospheric aerosol layer is dominated by explosive volcanic eruptions, injecting large amounts of $SO_2$
directly into the stratosphere. In recent years, large wildfires have also had a significant effect on the stratosphere (Ansmann et
al., 2018; Ohneiser et al., 2022; Thomason et al., 2023). Through their emissions, volcanic eruptions and wildfires can strongly
change the number concentration as well as the sizes of stratospheric aerosol. This is of great importance, since the chemical
as well as the radiative effects of the aerosol particles depend on their particle size distribution (PSD). This is why we need to
understand the variability of stratospheric aerosol size, especially after volcanic eruptions (Robock, 2015), and how long any
perturbations can be expected to last. There are multiple methods to investigate the size of stratospheric aerosols, each with
their own advantages and disadvantages. In situ measurements, e.g. balloon-borne or by aircraft, provide the most reliable char-
acterization of both the composition and PSD of stratospheric aerosol. A high resolution is possible both in terms of altitude
and in terms of particle radius. However, they can only provide sparse temporal and spatial sampling of the atmosphere.

This is where remote sensing satellite measurements come in. With satellite instruments it is possible to achieve a near
global coverage, depending on orbit and measurement geometry. This is important for the investigation of trends and seasonal
variability of stratospheric aerosols, for the observation of large scale perturbations caused by volcanic eruptions and wildfires
and for the validation of, e.g., climate models. However, there is only a very limited amount of satellite missions suited for high
quality observations of stratospheric aerosols. And again, each of these missions has its own strengths and weaknesses. Limb
scattering instruments, such as OMPS-LP (Flynn et al., 2014; Taha et al., 2020; Rozanov et al., 2024), OSIRIS (Llewellyn
et al., 2004; Bourassa et al., 2008) and SCIAMACHY (Bovensmann et al., 1999; Pohl et al., 2024), have the advantage of a
high measurement frequency and of usually achieving an almost global coverage daily. But they suffer from the necessity to
make assumptions about the stratospheric aerosol size distribution before being able to retrieve information about the aerosol
extinction and the particle size, as well as from the complicated light paths that have to be considered in radiative transfer
models and resulting issues with, for example, the surface albedo and differing data quality depending on the scattering angle.

Satellite solar occultation measurements, where the attenuation of the solar radiation is measured by looking through the
atmosphere and into the sun, have much less frequent opportunities to perform measurements, since those are tied to the orbit of
the instrument and the setting and rising of the sun behind the atmosphere from the point of view of the instrument. This comes
down to only a few tens of profiles measured per day. However, solar occultation measurements have a couple of advantages,
that make them especially well suited for the investigation of stratospheric aerosol size: The high signal strength of the sun
facilitates a small field of view, which results in a good vertical resolution being possible. The measurements are inherently
self-calibrating, since a measurement of the solar spectrum unaffected by the atmosphere can be taken each orbit, minimizing
the problem of longterm drift in the data over the time of the instrument's mission. Finally, the solar occultation geometry
facilitates a straight forward way to retrieve the aerosol extinction, without assumptions about the aerosol PSD being necessary
(McCormick, 1987).





In this work, the variability of the stratospheric aerosol size is investigated using the solar occultation data set of the Stratospheric Aerosol and Gas Experiment III on the Russian satellite Meteor-3M, in short SAGE III/M3M. With its measurements between 2002 and 2005 the SAGE III/M3M mission covers a valuable time frame. This is because the years after 2000 were

the only time frame to date within the satellite measurement era, i.e. since the late 1970s, that was close to background conditions and was characterized by relatively steady aerosol extinction levels (Deshler et al., 2006; Thomason et al., 2008). For the ongoing successor mission to SAGE III/M3M, SAGE III/ISS, which started in 2017, the stratosphere has been characterized by a state of perpetual perturbation due to different volcanic and wildfire events. The predecessor to SAGE III/M3M on the other hand, SAGE II, which covered the long time frame between 1984 and 2005, did indeed cover some volcanically more quiescent

periods. However, the much narrower spectral range of SAGE II restricted the possibility of aerosol size retrievals to the point of including unavoidable ambiguities, as discussed in Wrana et al. (2023). The wider spectral range of the SAGE III/M3M measurements on the other hand facilitates the application of the retrieval algorithm described in Wrana et al. (2021) (see Sect. 2.2), allowing for a more robust characterization of stratospheric aerosol size during background conditions. Additionally, high latitudes that are otherwise not often covered are observed by the SAGE III/M3M measurements, especially in the northern

hemisphere.

After introducing the main instruments and methods in Sect. 2, the evolution of the stratospheric aerosol size between 2002 and 2005 as well as some phenomena related to the high winter latitudes will be discussed in Sect. 3. Then, after the retrieval data set of this work is compared to in situ data in Sect. 4, an issue regarding the assumed PSD shape, that is relevant for the

presented data set as well as for remote sensing aerosol size retrievals in general will be discussed in Sect. 5.

## 2   Instruments and Methodology

### 2.1   SAGE III/M3M instrument

The SAGE III/M3M instrument, was launched in December 2001 and was operating and providing measurement data between February 2002 and November 2005. It is the successor of the SAM II, SAGE I and SAGE II satellite instruments and was

later followed by the SAGE III/ISS instrument. SAGE III/M3M gathered information on different atmospheric constituents like atmospheric water vapor, $O_3$, $NO_2$ and stratospheric aerosols.

SAGE III/M3M performed solar occultation measurements, i.e. it measured the attenuation of solar radiation due to atmospheric constituents by looking through the atmosphere into the sun during a sunrise or sunset event from the point of view of the satellite. Mounted on the Meteor-3M weather satellite it was in a sun synchronous orbit with an inclination of $99.5°$

(Roberts et al., 1996). Due to the platform's orbit around 15 sunrise and 15 sunset events were observed by the instrument in 24 hours and the sunrise and sunset measurements end up happening in different hemispheres, as depicted in Fig. 1. Sunrise and sunset is here referring to the sun rising or setting behind the earth from the point of view of the satellite during its orbit. The sunrise measurements oscillate roughly between $35°$ S and $60°$ S, while the sunset measurements cover the northern




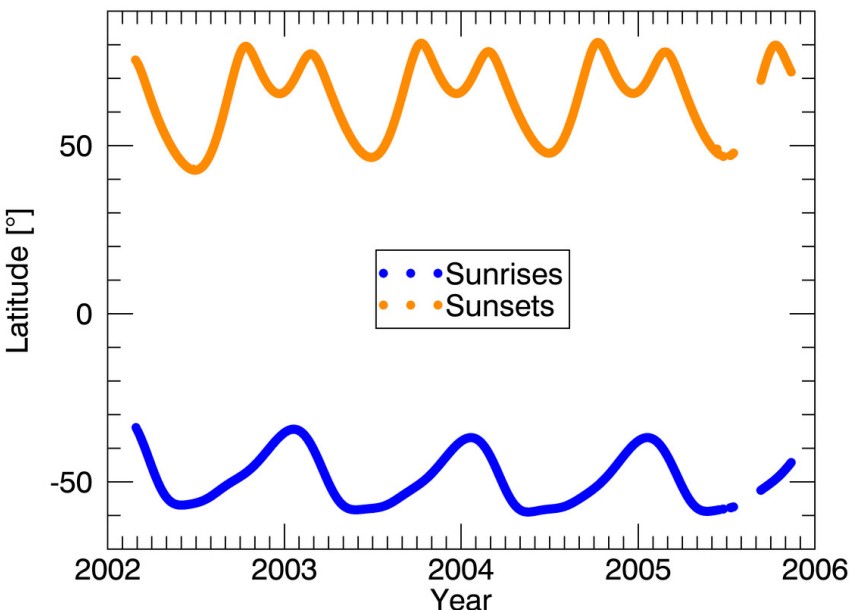

**Figure 1.** Coverage of geographical latitude by the SAGE III/M3M solar occultation measurements over the course of its mission. Sunrise events (blue dots) were only observed in the southern hemisphere, while sunset events (red dots) were observed in the northern hemisphere.

latitudes between approximately $40\,^\circ$ N and $80\,^\circ$ N. Since the scattering angle does not change between northern and southern
hemisphere solar occultation measurements, the data quality of both is likely very similar.

For a solar occultation measurement a scanning mirror was used to follow the sun's perceived ascent or descent from or towards the horizon as well as to scan across the sun's surface and produce multiple samples at each tangent altitude. After passing the neutral density filter and the science aperture, which defines the instantaneous field of view (IFOV) of the instrument, the solar radiation reaches the grating spectrometer and the detectors. Those consist of a CCD array covering the
wavelengths 280 nm to 1040 nm with a spectral resolution of 1 to 2 nm and an Indium Gallium Arsenide (InGaAs) infrared photodiode around 1550 nm with a 30 nm bandwidth (SAGE III ATDB, 2002; SAGE III/M3M User's Guide, 2004). This setup results in the SAGE III/M3M data set providing aerosol extinction coefficients between 0 and 40 km altitude in 0.5 km steps for 9 different wavelength channels between 384 and 1545 nm. This is the main data set used for the stratospheric aerosol size retrieval of this work. Version 4 of the SAGE III/M3M level 2 solar occultation data is used.
Thomason et al. (2010) evaluated the aerosol extinction coefficient channels of SAGE III/M3M based on comparisons to other satellite instruments as well as based on internal consistency. The channels at 449, 520, 755, 869 and 1021 nm are assessed to be reliable and have accuracies and precisions of around 10 % between 15 and 25 km. The same applies to the 1545 nm channel although a bias below 15 km cannot be excluded. The 385 nm channel is only recommended for use above



16 km altitude. The 601 nm has a much lower precision (~20 %) due to the influence of ozone absorption and it is generally
recommended to not use the 676 nm extinction coefficients. In this work, the channels 449, 755 and 1545 nm are used for the
stratospheric aerosol size parameter retrieval, as will be explained in more detail in Sect. 2.2.

## 2.2 Aerosol size retrieval method

The SAGE III/M3M solar occultation data set is used in this work to derive parameters describing the size of stratospheric
aerosol particles. This is done by comparing the aerosol extinction data provided in the measurement data set with a lookup
table, that was calculated using Mie theory (Mie, 1908). The method is in essence the same as the one that was also used in
Wrana et al. (2023) and is described in detail in Wrana et al. (2021).

A number of assumptions underly the retrieval method and therefore also have to be kept in mind, when interpreting the
results later on. Firstly, it is assumed, that the PSD of stratospheric aerosol can be described well by a monomodal lognormal
size distribution, which is defined as follows:

$$n(r) = \frac{dN(r)}{dr} = \frac{N_0}{\sqrt{2\pi} \cdot r \cdot ln\sigma} \cdot exp\left(-\frac{ln^2(r/r_{med})}{2ln^2\sigma}\right) \tag{1}$$

Here, the three parameters controlling the monomodal lognormal size distribution are the median radius $r_{med}$, the mode
width $\sigma$ and the total number density $N_0$. This monomodality is a common assumption in stratospheric aerosol size retrievals
from satellite measurements, on the one hand because it can often be a good model of the true conditions, but more importantly
out of necessity, since usually there is not enough independent spectral information in satellite measurements to facilitate a
retrieval with a more complex model, e.g. a bimodal lognormal distribution. Still, a monomodal lognormal distribution can be
inadequate in some cases. The effect this would have on the size retrieval of this work is described in Sect. 5.

Furthermore it is assumed, that the individual droplets of the stratospheric aerosol are spherical, which is a prerequisite for
Mie theory to be applicable. Since stratospheric aerosols are mostly liquid droplets of sulfuric acid and water with a very small
size, usually well below 2 μm (Sugita et al., 1999; Deshler et al., 2003b; Deshler, 2008; Li et al., 2023), this is expected to be
very close to reality.

The basis of the aerosol size retrieval is the lookup table that is calculated using a Mie Code (Mie scattering routines,
2018). It consists of aerosol extinction ratios at the wavelengths 449 nm / 755 nm and 1545 nm / 755 nm, both calculated for
many combinations of different theoretically plausible values of the median radius and the mode width $\sigma$. The value range
covered this way by the lookup table is 1 nm to 1000 nm for the median radius and 1.05 to 2.0 for the mode width. The real
and imaginary refractive indices necessary for the Mie calculations are taken from Palmer and Williams (1975) and adjusted
for temperature using Lorentz-Lorenz-corrections as described by Steele and Hamill (1981). The three wavelengths that are
used by the two extinction ratios are chosen to match three of the spectral channels of the SAGE III/M3M instrument, at
which aerosol extinction coefficients are provided. These SAGE III/M3M spectral channels (449 nm, 755 nm, 1545 nm) in
turn were chosen because they make use of the wide spectral range of the instrument while also only including reliable aerosol



extinction channels (Thomason et al., 2010). Now that the lookup table is prepared, the median radius and mode width value can be retrieved from a given set of extinction ratios from the measurements via interpolation of the lookup table values. More detailed information on the size retrieval approach was provided in Wrana et al. (2021).

The retrieval of median radius and mode width is independent of the total Number density $N_0$, because the latter cancels out
when forming the extinction ratios. The number density can be calculated after obtaining $r_{med}$ and $\sigma$, through the following relation:

$$N_0 = \frac{k_{ext}(\lambda)}{\sigma_{ext}(\lambda)} \tag{2}$$

Here, $k_{ext}(\lambda)$ is the measured extinction coefficient and $\sigma_{ext}(\lambda)$ is the extinction cross section at wavelength $\lambda$ simulated from the retrieved median radius and mode width values.

Another useful quantity, that can be calculated from the retrieval results is the effective radius, which is the area weighted mean radius (Grainger, 2023). Since larger aerosol particles scatter light more strongly in the relevant size range it is more indicative of the part of the PSD that is responsible for most of the radiative effect of the stratospheric aerosol. It can be calculated from $r_{med}$ and $\sigma$, or from the aerosol volume density $V$ and surface area density $A$ in the following way:

$$r_{eff} = r_{med} \cdot exp\left(\frac{5}{2} \cdot ln^2(\sigma)\right) = \frac{3V}{A} \tag{3}$$

Lastly, the absolute mode width $\omega$, as introduced by Malinina et al. (2018), is a very useful parameter to understand how wide the PSD is in linear radius space. It is much more useful than the mode width $\sigma$, which only provides information about the width of the size distribution in logarithmic radius space and can therefore easily be misinterpreted. This is illustrated and explained in more detail in Wrana et al. (2023). The absolute mode width $\omega$ is calculated as follows:

$$\omega = \sqrt{r_{med}^2 \cdot exp(ln^2(\sigma)) \cdot (exp(ln^2(\sigma)) - 1)} \tag{4}$$

Additionally, an accuracy parameter has been defined to exclude noisy data in the retrieved quantities. It relates the lookup table values to the propagated uncertainties of the extinction ratios and is described by Wrana et al. (2021). For all retrieval data shown here, data points with respective accuracy parameter values below 16 are excluded, as it was done in Wrana et al. (2021, 2023) before. Over the whole data set this filters out 35.9 % of the retrieved values overall, but much less in the maximum of the Junge layer, e.g. only 11 % of data at the 18 km altitude level.

## 2.3   SAGE II instrument

SAGE II (Stratospheric Aerosol and Gas experiment II) was the predecessor to SAGE III/M3M. It was mounted on the Earth Radiation Budget Satellite (ERBS). Its mission spanned from October 1984 to August 2005, i.e. with considerable overlap



with the SAGE III/M3M mission. The platform was in an orbit at 610 km altitude with an inclination of 57° (McCormick, 1987; Thomason et al., 2008).

Like SAGE III/M3M, SAGE II also measured stratospheric aerosol, ozone, $NO_2$ and water vapor through the use of the solar occultation geometry. Aerosol extinction coefficients are provided with spectral channels at 386, 452, 525 and 1020 nm in 0.5 km steps between the Earth's surface and 40 km altitude. The aerosol extinction profiles were found to be reliable and robust during the time period covered by SAGE III/M3M with a bias lower than 20 % at 525 nm below 25 km altitude (Thomason et al., 2008). Due to the orbit parameters and the solar occultation geometry, roughly 30 profiles were measured per day. The instrument covered all latitudes between 80°N and 80°S, with a gradual daily shift in latitude of the sunrise and sunset measurements, respectively. In this work, version 7.0 of the SAGE II data in binary format (NASA/LARC/SD/ASDC, 2012) was used.

### 2.4  Kiruna OPC measurements

Using an optical particle counter (OPC), the size of individual aerosol particles can be measured. This is a major advantage over satellite measurements, where the measured signal originates from many different aerosol sizes. The University of Wyoming carried out balloon-borne in situ measurements between 1990 and 2004 from Kiruna (68°N, 21°E), Sweden. Both the time frame and the location of the measurements open up the possibility of comparisons with the high northern latitude sunset measurements of SAGE III/M3M.

The in situ measurements in Kiruna were carried out with the Wyoming white light optical particle counter (WOPC) by the University of Wyoming. The instrument measured the size of aerosol particles with a radius of 150 nm or larger in up to 12 size classes. For this, the scattering of white light by, ideally, single aerosol particles is measured at an angle of 40° relative to the incident light of the incandescent lamp. The scattered light was focused onto a pair of photomultiplier tubes (PMTs) for pulse height detection. To reduce noise, only coincident PMT pulses of two independent but symmetrical photon paths were counted towards the integral particle concentrations. Air was guided through the inlet with a flow rate of 10 L/min. Aerosol size is retrieved from the measurements through the use of Mie Theory and an assumed refractive index of 1.45, assuming sulfate aerosol particles. Also included are separate measurements of the total number of condensation nuclei (CN) with a size larger than 10 nm. This is achieved through the use of ethylene glycole vapor and a growth chamber, forcing the aerosol particles to grow to a detectable size through condensation of ethylene glycol. This effectively gives a measure of the total aerosol number density (Deshler and Oltmans, 1998; Norgren et al., 2024). By coupling the total aerosol number to the WOPC measurements, the measurements can be fitted with an assumed PSD to provide a means of comparison with satellite measurements that retrieve a smaller number of parameters. For this the WOPC data are fitted with both unimodal and bimodal lognormal PSDs (Deshler et al. , 2019). For this work the unimodal, or monomodal, data of UWv2.0 is used (Deshler, 2023).





## 3 Retrieval results

### 3.1 Temporal evolution of stratospheric aerosol size

In Figs. 2 and 3 the temporal evolution of some of the retrieved stratospheric aerosol particle size distribution parameters is illustrated. Fig. 2 depicts the data from the northern hemisphere, i.e. the sunset measurements, while Fig. 3 corresponds to the southern hemisphere and the sunrise measurements. The data shown are daily zonal mean profiles combined over the whole SAGE III/M3M mission time span from February 27th, 2002 to November 12th, 2005. The topmost plot shows the mean latitude of each day to which the profiles of the color plots correspond. The retrieved parameters shown in the color plots are,

from top to bottom, the median radius, the absolute mode width $\omega$, the effective radius and the number density.

The oscillation of the vertical extent of the aerosol layer over time in the color plots is in large part explained by the latitudinal oscillation of the SAGE III/M3M measurements (see Fig. 1 or topmost plots of Figs. 2 and 3). A red line in each color plot shows the tropopause height, which is provided in the SAGE III/M3M data. The range of values for each parameter found in the entire SAGE III/M3M data set is indicated by the color bar to the right of each plot. Some outliers have been excluded from

the color plots due to their infrequency and the consequent widening of the displayed range of values, which would obscure the important signatures and structures within the plot. Values directly above the tropopause are often missing due to the filtering of data using the accuracy parameter described in Sect. 2.2.

Vertical dashed lines mark the times of volcanic eruptions that were strong enough to potentially have had an effect on the stratospheric aerosol layer. The first two were small but significant eruptions by two tropical volcanoes in late 2002. Namely,

Ruang (2.3 °N) on September 25th, 2002 (Global Volcanism Program, 2023) and Reventador (0.08 °S) on November 3rd, 2002 (Hall et al., 2004). According to the Darwin Volcanic Ash Advisory Center, the plume of Ruang was rising up to 17 km (Global Volcanism Program, 2023). The Reventador plume reached 17 km as well and emitted 0.06 - 0.08 Tg $SO_2$ (Hall et al., 2004; Global Volcanism Program, 2023). The third dashed vertical line marks the peak of the eruptive phase of the Manam volcano (4.08 °S) that lasted from October 2004 until the end of January 2005. The peak occurred on January 27th-28th, 2005, with the

volcanic plume reaching into the stratosphere, up to 21-24 km altitude and emitting approximately 0.14 Tg of $SO_2$ (Tupper et al., 2007; Global Volcanism Program, 2023). It should be mentioned, that prior to this, between October 24th and November 27th 2004, Manam also emitted around 0.15 Tg of $SO_2$ at altitudes between 6 and 18 km making it unclear how much of it reached and impacted the stratosphere before the peak in January 2005 (Global Volcanism Program, 2023).

In the northern Hemisphere (Fig. 2), the strongest signatures in the SAGE III/M3M time series for each parameter are found

in the months after the three tropical volcanic eruptions. Without necessarily claiming causality, the median radius, absolute mode width $\omega$ and the effective radius show decreased values starting two to three months after the volcanic eruptions in the lower Junge layer and lasting for over half a year, as indicated by the darker colors. In contrast, the number density, which is shown with a logarithmic color scale, shows a very strong increase in the same regions. In the context of the assumed monomodal lognormal shape of the PSD this means, that its peak is shifted towards smaller radii and its width is reduced, i.e.

smaller particles than before dominate the optical signal measured by the instrument. The effect of the monomodal assumption is discussed further in Sect. 5. The observed overall pattern in the retrieved parameters is strikingly similar to the results shown



**Figure 2.** Northern hemisphere (i.e. from sunset occultation events): Daily averages of PSD parameters, retrieved from the SAGE III/M3M stratospheric aerosol extinction data. From top to bottom: Latitude coverage, median radius, absolute mode width $\omega$, effective radius, number density. The red line indicates the tropopause height. Note that the number density values are given on a logarithmic scale.



**Figure 3.** Southern hemisphere (i.e. from sunrise occultation events): Daily averages of PSD parameters, retrieved from the SAGE III/M3M stratospheric aerosol extinction data set. The red line indicates the tropopause height.



in Wrana et al. (2023), where eruptions of the Ambae, Ulawun and La Soufrière volcanoes between 2018 and 2021 where shown to also having led to a decrease of median radius, absolute mode width and effective radius and to a strong increase in number density. This was backed up by both remote sensing retrieval data and model simulations, which generally showed a good agreement. Similar to Ruang, Reventador and Manam, these were all smaller volcanic eruptions, i.e. eruptions emitting less than $1\,\mathrm{Tg}$ of $SO_2$ into the stratosphere. Also, all of these eruptions were tropical. Both the emission strength and the latitudinal location of the volcano could be important causal factors for the observed reduction in average aerosol size (Wrana et al., 2023). The low temperatures in the tropical upper troposphere/lower stratosphere (UTLS) region may be especially important due to their strong positive effect on nucleation of new particles from gaseous $H_2SO_4$ (Vehkamäki et al., 2002; Korhonen et al., 2003) formed from the emitted $SO_2$. The results of model simulations indicate that lower temperatures result in a shift in the aerosol size distribution towards lower radii (Pirjola et al., 1999).

Another notable signature found after the volcanic eruptions, particularly in the northern hemisphere, is an increase in average aerosol size in a roughly five kilometer thick layer above the discussed layer of reduced aerosol size. This increase is seen in the median radius and effective radius, but not necessarily in the absolute mode width $\omega$, i.e. the retrieved monomodal lognormal PSD remains at a similar width but its peak shifts towards larger radii. This size increase is much more pronounced in 2005 after the Manam eruption.

In the retrieval parameters of the southern hemisphere, which are shown in Fig. 3, a size reduction after the Ruang and Reventador eruptions can be seen as well. Both the increase in number density and the reduction in median radius and effective radius are much weaker than in the northern hemisphere. However, the signals are visible earlier here, which makes sense, since the measurements in the southern hemisphere were taken at lower latitudes, i.e. closer to the tropical volcanoes, than in the northern hemisphere. Increased aerosol size, best visible in the median radius and effective radius, and decreased number densities are found throughout the year 2004. The strongest signals in the southern hemisphere are the increased number density and the reduced median radius in the lowermost four to five kilometers of the Junge layer throughout the year 2005 and starting around the eruption of Manam.

## 3.2 Origin of aerosol size change

The SAGE III/M3M measurements did not cover the tropics and lower latitudes, as shown in Fig. 1. Therefore we have to look at the question whether the observed aerosol size reduction signals can plausibly be traced back to the tropical volcanic eruptions of Ruang, Reventador and Manam, or if other causes have to be considered.

In terms of the chronological sequence it seems plausible, that these signatures can indeed be of volcanic origin, since the strongest size reduction signals are found a few months after the eruption dates, i.e. there was enough time for the plumes to be transported from their possible tropical origin to the observed higher latitudes. However, there may also have been non-volcanic factors influencing the evolution of aerosol size and leading to an average size reduction in 2003 and 2005. An important factor may be the seasonally occurring new particle formation events in the polar winter stratosphere (Hofmann, 1990; Wilson et al., 1990; Campbell and Deshler, 2014), also known as the springtime condensation nuclei (CN) layer, that is linked to the onset of photolysis in polar spring (Mills et al., 2005).



An additional phenomenon and possibly also linked to the polar springtime CN layer are meteoric smoke particles (MSPs) that are seasonally accumulating in the polar winter stratosphere (Megner et al., 2008). They originate from meteoroids continuously entering Earth's atmosphere and being vaporized in the upper mesosphere and lower thermosphere in a process called meteoric ablation. Through this, layers of neutral metal atoms and ions are formed. The vaporized material recondenses to

form the meteoric smoke particles (Hunten et al., 1980; Plane, 2012), which by coagulation likely grow up to sizes of 40 nm (Megner et al., 2008; Bardeen et al., 2008). As part of the general atmospheric circulation the MSPs in the mesosphere are transported to the winter hemisphere and then downwards in the polar vortex (Megner et al., 2008). Through the lowering of the $H_2SO_4$ saturation vapor pressure due to the combination with meteoric material, sulfuric acid is suggested to be able to condense above the usual confines of the Junge layer (Saunders et al., 2012; Hervig et al., 2017). This way meteoric-sulfate

aerosols may be formed, that coagulate and grow into measurable sizes during their continued descent in the stratosphere (Schneider et al., 2021). The signal of low values in e.g. the median radius and effective radius that seemingly descends from the upper end of the Junge layer in January 2003 (see Fig. 2) could theoretically be related to this.

These seasonal nucleation/condensation events that include the MSPs accumulating in the polar winter stratosphere could be an important factor in the observed size evolution and reduction of average aerosol size in northern hemispheric winter. If

that were the case, there would be an unknown error in the retrieved aerosol size due to the refractive indices used in the Mie calculations of the lookup table, that are based on the assumption of pure sulfate aerosols. Unfortunately, little is known about the actual properties of meteoric smoke particles, mainly because of the difficulties of performing in situ measurements in such high altitudes (Megner et al., 2008).

However, the size reduction signal is visible well before spring, especially in 2003. Also, in particular after the Manam erup-

tion, a size reduction signal is found in the southern hemisphere as well, albeit weaker, i.e. in the summer hemisphere where no seasonal nucleation event would be expected. Adding to this point, the southern hemisphere measurements were taken at relatively low latitudes. Additionally, no comparable size reduction is found in 2004, when notable volcanic eruptions strong enough to influence the stratosphere were absent, except for a very weak possible signal in the southern hemisphere around January and February 2004. These points argue for the signals being at least in part of volcanic origin.


As an additional piece of information, the measurements of the SAGE II instrument, the predecessor of SAGE III/M3M, can be used to shed some light on the temporal and spatial dispersion of the volcanic plumes. Its mission started in 1984 and ended in August 2005, therefore temporally covering most of SAGE III/M3M's mission. The reason for the additional information gained on the volcanic plume dispersion by using the SAGE II instrument's data is its different orbit. Because of it a wider

latitude range is covered, between up to 80 °N and 80 °S, depending on season. The tropics are included as well, facilitating the tracking of the volcanic plumes, albeit with a relatively poor temporal resolution due to the sampling of solar occultation measurements.

Monthly zonal means of the aerosol extinction coefficient at 525 nm, as provided in the SAGE II v7.0 data set, are shown in Fig. 4 and the top row of Fig. 5. Extinction coefficients with associated relative uncertainties of 30 % or higher were excluded.

The extinction data was averaged onto a latitude grid with 5° bin size and plotted logarithmically. Even though monthly





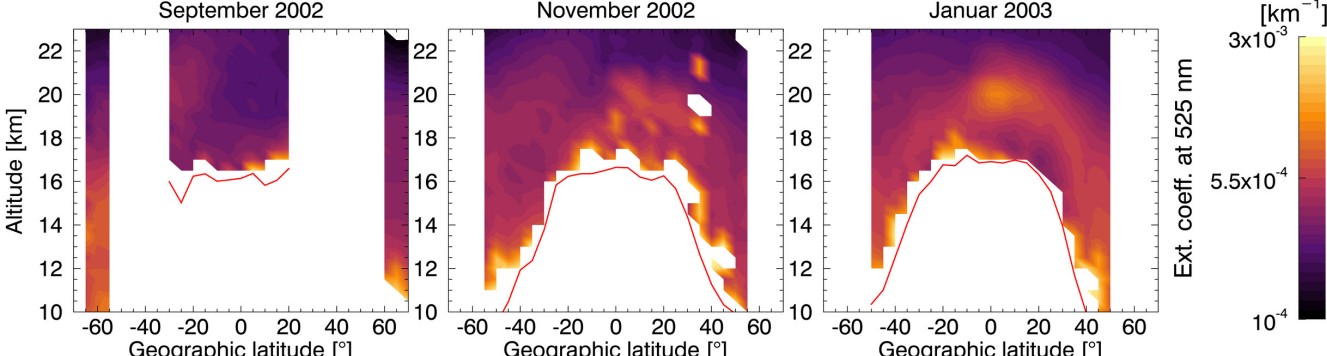

**Figure 4.** Ruang (2.3 °N) and Reventador (0.08 °S) plume dispersion indicated via monthly averages of the aerosol extinction coefficient at 525 nm as provided in the SAGE II v7.0 data set (NASA/LARC/SD/ASDC, 2012). Shown are characteristic months before (first panel) and after the volcanic eruptions. The red line indicates the tropopause height, data below it is excluded.

averages were calculated there are still data gaps, which is mostly due to the sparse sampling of satellite solar occultation measurements. The sampling of SAGE II also leads to the fact that in the shown plots the data from different latitude bins can come from different days within the month that the data is averaged over. Both figures show characteristic months around the volcanic eruptions in question, i.e. Ruang and Reventador for Fig. 4 and Manam for Fig. 5. In both cases, the leftmost

plot shows the atmospheric conditions before the respective volcanic eruptions, while the other two show the emergence and dispersion of the aerosol extinction enhancement due to the volcanic plume.

Before the Ruang and Reventador eruptions, which happened on September 25th and November 3rd, respectively, there are data gaps preventing a good look at the prevolcanic conditions. The best overview is possible in September 2002, first panel of Fig. 4, where at least some high northern latitude measurements are included. From this to the other panels for November 2002

and January 2003 a clear increase in the extinction coefficient can be seen in the Junge layer between the tropics, where the volcanoes are located, and the northern mid-latitudes, where the SAGE III/M3M measurements happened. It can be concluded, that the size reduction signal found in early 2003 in the northern hemisphere (see Fig. 2) is very likely connected to the volcanic eruptions of Ruang and Reventador in late 2002.

For Manam the picture is not as clear. In the top row of plots in Fig. 5, a strong and clear increase in aerosol extinction

coefficient is visible in the tropics after the Manam eruption in March 2005. However, whether the volcanic plume reached higher latitudes or not is not as easily discernible. For this reason an additional plot was added to Fig. 5. It shows the change in extinction coefficient at 525 nm between December 2004 and March 2005, i.e. the difference between the first and third plot of the top row, in percent. To distinguish it from the extinction coefficient data itself another color scheme is used. The data is presented logarithmically. The plot suggests at least some transport of the Manam volcanic plume from the tropics

towards mid-latitudes. Transport to the southern hemisphere shows up as a distinct enhancement of the typical Junge layer, with increases between roughly 10 and 30 %. In the northern hemisphere, however, an increase of a similar magnitude can be seen not in a layered shape but throughout the lower stratosphere. Especially since this time frame covers mostly the northern





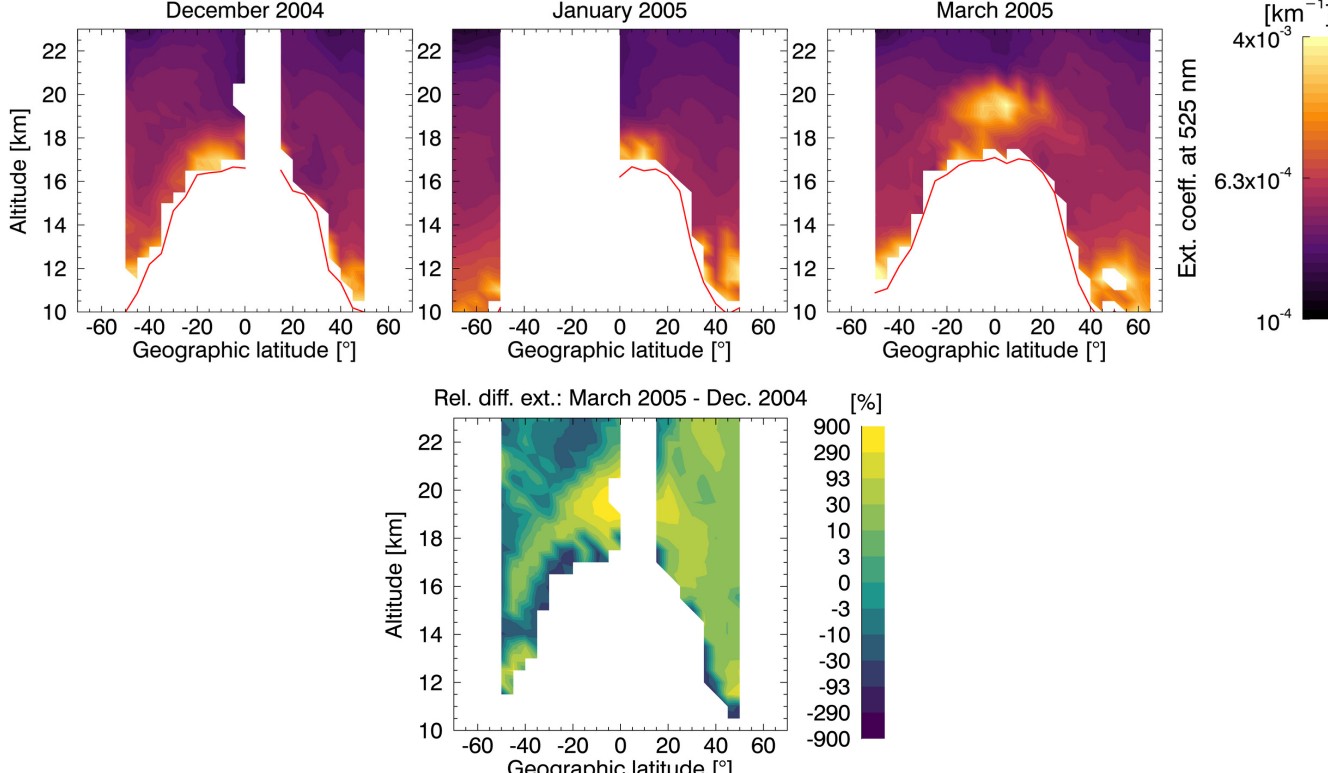

**Figure 5.** Top row of plots: Manam (4.08 °S) plume dispersion indicated via monthly averages of the aerosol extinction coefficient at 525 nm as provided in the SAGE II v7.0 data set (NASA/LARC/SD/ASDC, 2012). Shown are characteristic months before (first panel) and after the volcanic eruption. The red line indicates the tropopause height, data below it is excluded. Lower plot: Relative difference in percent between the extinction coefficients in March 2005 and December 2004 (i.e. between top rightmost and leftmost plots).

hemispheric winter, a strong seasonal component may be at play here, with increasing aerosol extinction coefficients due to a lowering of the temperature. There also is a cluster of high aerosol extinction coefficient values (first row of plots) in the low-
ermost stratosphere between roughly 40 °N and 60 °N, that exists even in December 2004, before the main volcanic eruption on January 27th-28th 2005. This cluster of values is, however, associated with high relative uncertainties and may therefore simply be a retrieval artifact rather than a real signal. It therefore is challenging to determine from the SAGE II data to what degree the aerosol size reduction in the northern hemisphere after the Manam eruption (see Fig. 2) is attributable to the eruption or to seasonal variation, or to a combination of both.


Taking everything discussed above into account it is likely, that the average aerosol size reduction signals that were observed after the Ruang and Reventador eruptions in both hemispheres and after the Manam eruption in at least the southern hemisphere are indeed volcanic effects. If that is the case, these volcanic events may be comparable to the ones discussed in Wrana et al. (2023). In case the signals are indeed of volcanic origin, the fact that the size reduction signals are stronger in the northern



hemisphere (Fig. 2) than in the southern hemisphere (Fig. 3) can likely be explained by the tropical location of the volcanoes, the timing of the eruptions in northern winter and the tendency of the Brewer-Dobson-Circulation of enhanced transport from the tropics towards the winter hemisphere (Kremser et al., 2016).

### 3.3    Possible polar stratospheric cloud (PSC) observations

Polar stratospheric clouds (PSCs) can form in the lower stratosphere and play an important role in both the activation of halogen
species resulting in ozone destruction and the ozone hole and in the denitrification and dehydration of certain regions of the polar stratosphere (Tolbert and Toon, 2001; Lowe and MacKenzie, 2008). Since the stratosphere is typically very dry, the very low temperatures in the stratospheric polar winter are a precondition for them to form (Peter, 1997). There are multiple known types of PSC, whose particles have different compositions and layers and which can be liquid or frozen, based on the conditions in which they form (Lowe and MacKenzie, 2008; Tritscher et al., 2021).

There can be solid PSC particles, such as nitric acid trihydrate (NAT) (Voigt et al., 2000; Hanson and Mauersberger, 1988) and ice particles, as well as liquid supercooled ternary solution (STS) droplets (Carslaw et al., 1994; Tabazadeh et al., 1994; Deshler et al., 2003a), which are the most commonly occuring type of particle in a PSC. Also, liquid and solid particle types can often coexist (Weisser et al., 2006). The formation of these different kinds of PSC particles depends foremost on temperature, with NAT particles having the highest equilibrium temperature. At temperatures roughly 3 K above the frost point, STS particles
form. Ice particles then nucleate 3 to 4 K below the frost point (Weisser et al., 2006). Other factors, like the preexistence of other particles may also play a role. For example, ice particles can provide surfaces for the heterogeneous nucleation of NAT particles (Koop et al., 1997) and meteoric particles may play a role in the heterogeneous nucleation of both NAT and ice particles (Bogdan et al., 2003; Tritscher et al., 2021).

Since the SAGE III/M3M measurements in part covered high latitudes, mostly in the northern hemisphere, there was a
potential to encounter polar stratospheric clouds (PSCs) in the data set. In Fig. 6, the top plot again shows the latitudinal coverage of the measurements in the northern hemisphere. Below this, time series of the aerosol extinction coefficient at 449 nm and the temperature are shown, both parameters being provided in the SAGE III/M3M data set. For visibility of the relevant signals, the extinction coefficient is plotted with a reverse color scale, i.e. darker colors in this case mean higher values. To facilitate referencing of the same areas in the different plots, extinction coefficient values are only shown where aerosol size
parameters could be retrieved and temperature values are only plotted where extinction coefficients are shown as well.

The two main areas of interest are immediately obvious, when looking at the extinction coefficient plot: In the winters of 2002/2003 and 2004/2005 there are multiple clusters of only a few days of perturbed aerosol extinction at the uppermost altitudes of the Junge layer (and in part above it) including some of the highest values of the whole time series. The short duration of the signals as well as the position at the uppermost end of the aerosol layer basically rule out a volcanic origin.
Looking at the lowermost plot, both winters included the lowest temperatures over the longest times in the SAGE III/M3M time series. More specifically, the clusters of high extinction coefficient in question also exhibit very low temperatures, well below 200 K, with the lowest temperature in these daily average profiles being 186 K. It therefore is very likely, that the



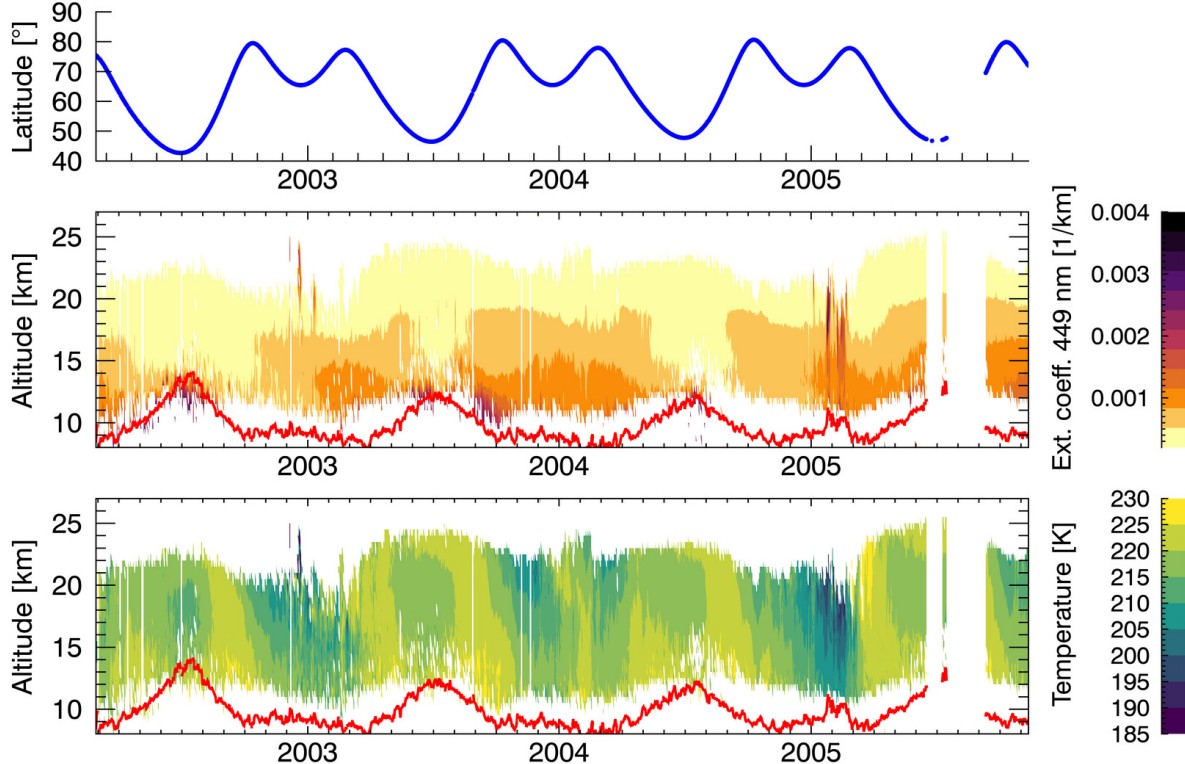

**Figure 6.** Daily averages of parameters, that are related to potential PSC observations in the SAGE III/M3M data set, in the northern hemisphere (i.e. from sunset occultation events). From top to bottom: Latitude coverage, aerosol extinction coefficient at 449 nm, Temperature, all provided in the SAGE III/M3M data set.

observed sporadic enhancements in the extinction coefficients come from PSCs, that had been forming due to the cold winters in 2002/2003 and 2004/2005.

In the cases that the aerosol size retrieval method described in this work does retrieve results for the measurements containing a PSC signal, median radii of up to roughly 350 nm are found. However, these results are in no way reliable. Firstly, the actual refractive indices of the PSC particles may differ more or less strongly from the values assumed for pure sulfate aerosols. Secondly, the particles may not even be liquid and therefore not spherical, in which case Mie theory would not even apply. For probably the same reasons there are many measurements where the retrieval method does not work and produces no results at

all, since the extinction ratios formed from the measurements fall outside of the lookup table value range.

## 4    Comparison to OPC measurements

Since the OPC measurements can be assumed to be more reliable in determining PSD parameters than satellite retrieval data, in this section both are compared for validation of the retrieval data of this work. The only stratospheric OPC measurements



made using University of Wyoming aerosol counters in the southern hemisphere during the SAGE III/M3M mission were at
78 °S, i.e. too far south for a comparison with SAGE III/M3M. However, there have been OPC measurements in Kiruna (68 °N,
21 °E), Sweden, on eight days in total, most of them in December 2002.

In Fig. 7 a comparison of the extinction coefficient at 449 nm, the effective radius, the absolute mode width $\omega$ and the
aerosol number density from both the Kiruna OPC measurements and the retrieved data from collocated SAGE III/M3M
sunset oberservations is shown. Additionally, in the topmost plots the temperature as provided in the SAGE III/M3M (light
blue) and OPC data set (dark blue) is shown. Each column of plots shows data from one of three different days, i.e. December
3rd and 4th, 2002 and January 10th, 2004. Finding good collocations is difficult, since both the OPC and SAGE III/M3M
measurements are sparse in their own way. That is why the criteria for maximum spatial distance and temporal difference
between the in situ and satellite measurements had to be chosen case by case in order to find a balance between sufficient
sample size and actual comparability of the measured air volumes. For each date, three SAGE III/M3M events were found as
good collocations and averaged to obtain the shown profiles. The maximum distance between the OPC measurement location
and the collocated solar occultation events, in order of the three dates, was 1116 km, 760 km and 915 km with a maximum time
difference of 24 h. The balloon flight on January 10th, 2004 was actually targeted on a SAGE III/M3M overflight. The satellite
retrieval data of this very close sunset event is included but due to the noisy nature of individual profiles it is still averaged
with two other profiles in order to have more altitude levels with reliable data. The SAGE III/M3M extinction data has roughly
a 0.5 km vertical resolution and is also provided on a 0.5 km vertical grid (SAGE III/M3M User's Guide, 2004). To achieve
comparability, the OPC data which is provided on a finer grid is averaged to the same 0.5 km grid. The shown profiles of the
SAGE III/M3M PSD parameters cover a smaller altitude range than the OPC data, because the uncertainties of the extinction
coefficients and therefore the uncertainties of the extinction ratios which are used for the aerosol size retrieval are larger at the
lower and upper end of the Junge layer. This leads to measurements more often falling outside the value range of the lookup
table where no result can be retrieved and to more data being filtered out as unreliable through the accuracy parameter (see
Sect. 2.2). Averaging over a larger amount of measurements would yield more data here, but defeat the purpose of the data
set comparison. In general the comparison shows a good agreement between both retrieval data sets across the retrieved PSD
parameters, at least above roughly 15 km altitude.

Interestingly, the Kiruna WOPC measurements in December 2002 were targeted at PSCs that formed in the 2002/2003
arctic winter in a large scale low temperature region over the North Atlantic. Particularly on December 4th, 2002 multiple
passes through a PSC were achieved through controlled up and down motions of the balloon, which points towards the large
horizontal scale of the cloud. During the WOPC measurements of this event the PSC was found roughly between 20 and 27 km
and both liquid STS and solid particles, possibly NAT, were identified (Larsen et al., 2004; Weisser et al., 2006).

In the middle column of Fig. 7, the PSC signal can clearly be seen in both data sets in the extinction coefficient at 449 nm
above 22 km altitude, although the satellite data has higher values across all altitude levels. This is accompanied by temper-
atures below 190 K, with in part lower temperatures in the in situ data set than in the satellite data set, which points towards
air volumes with different conditions likely having been observed by the satellite and the in situ measurements. Above 22 km
altitude, in the PSC region, the size distribution parameters retrieved from SAGE III/M3M agree qualitatively with the OPC





values, both showing an increased effective radius and absolute mode width of the size distribution. The absolute mode width
$\omega$ shows a remarkable agreement above 17 km, the OPC effective radii however show much higher values than the satellite
retrievals in the PSC layer above 20 km. It should be noted, that both data sets assume aerosol particle compositions, and
therefore refractive indices, of typical pure sulfate particles, which may introduce considerable errors to the retrieval results.

Regarding the state of the Junge layer, below 20 km, both the December 3rd and 4th measurements fall more or less into the
category of background conditions, since the signals likely linked to Ruang and Reventador have not shown up yet at that point
and the time before has been mostly volcanically quiescent.

The comparison of January 10th, 2004 shows the overall best agreement between OPC and satellite retrievals. This result is
encouraging and perhaps not fortuitous. The balloon measurement on January 10th was made specifically for a comparison with
a SAGE III/M3M overpass of the balloon launching site and was coincident in time and spatially close. Also the temperatures
indicate there is little influence from PSCs, in contrast to the other two comparisons shown. The likely effects of Ruang and
Reventador on stratospheric aerosol size have waned up to this point. Fig. 6 though shows a clear perturbation of the extinction
coefficient starting in summer 2003 and continuing well into the following year. The source of this signal is not clear. It could
in part still be influenced by the volcanic material from the Ruang and Reventador eruptions, although another source seems
likely due to the timing of this new maximum in the time series. During the northern summer 2003, there have only been
small volcanic eruptions, such as the Anatahan (16.4 °N) eruption on May 10th-12th, whose ash plume rose up to 13.4 km, or
the Bezymianny (56 °N) eruption on July 26th 2003, where the emitted ∼0.003 Tg of $SO_2$ rose up to approximately 11 km
(Global Volcanism Program, 2023). In their analysis of smoke events from SAGE II data Thomason et al. (2023) identified
only two significant smoke events in 2003: One event in January in Canberra, Australia, i.e. in the southern hemisphere and
one in August at Conibear Lake, Canada, which had only a limited impact on the stratosphere after filtering out data based on
tropopause height.

Since the amount of available collocated measurements to analyze is so low, it is not possible to derive generalized conclu-
sions. Still, there may be a systematic difference between the OPC and SAGE III/M3M PSD parameters below roughly 15 km.
For all three days, the effective radius and $\omega$ values are higher and the number densities are lower in the satellite retrieval
data than in the in situ data, for these altitudes. Since both datasets being compared here share the same assumption on the
monomodal lognormal PSD shape and similar assumptions on the refractive indices it is currently not clear what the origin of
this discrepancy could be. One possible factor may however be the stated possibility of a positive bias in the 1545 nm chan-
nel of SAGE III/M3M below 15 km (Thomason et al., 2010). If such a positive bias existed, it would on average lead to an
overestimation of the average aerosol size and an underestimation of the particle number density.

Finally, it is difficult to say how much of the differences found are simply a result of the sampling of too different air volumes
due to the sparse spatial coverage of solar occultation measurements. In addition, the in situ measurements sample in a very
limited spatial region, while the SAGE III/M3M profiles shown are representative of averages over multiple measurements,
which themselves correspond to air volumes in the hundreds of cubic kilometers (SAGE III ATDB, 2002). Keeping this in
mind, the agreement between the two data sets is better than expected.





**Figure 7.** Comparison of PSD parameters retrieved from SAGE III/M3M (orange) and from OPC measurements in Kiruna (Deshler, 2023) (black) for three different days (columns). Shown are, from top to bottom, the aerosol extinction coefficient at 449 nm, the effective radius, the absolute mode width $\omega$ and the aerosol number density. Additionally, the first row of plots includes the average temperature as provided in the OPC data set (dark blue) and in the SAGE III/M3M data set (light blue). The red horizontal line indicates the average tropopause height for the SAGE III/M3M measurements.



## 5 Discussion of monomodality

The assumption of a monomodal lognormal shape of the stratospheric aerosol PSD is at the basis of both the SAGE III/M3M
and the OPC data set that were compared in Sect. 4. Even though the assumption is a necessity for the satellite data set due to
the limited amount of independent spectral information, its effects have to be evaluated critically.

Occultation measurements are sensitive to a directed bias of the retrieved size distribution parameters, if the assumed shape
of the PSD is wrong, e.g. if the actual size distribution is close to a bimodal lognormal distribution as opposed to the monomodal
lognormal distribution, which is assumed in this paper. This is relevant for all remote sensing geometries, but affects e.g. lidar
and occultation measurements differently because of their different sensitivities to smaller and larger aerosol particles (von
Savigny and Hoffmann, 2020).

Mie calculations and a test retrieval with synthetic data were conducted to illustrate how it would affect the aerosol size
retrieval presented in this work if the actual PSD were to deviate from the assumed monomodal lognormal shape. Fig. 8 a)
shows the reference case, where the true size distribution (blue curve) is a monomodal lognormal PSD with a median radius
of 150 nm and a mode width $\sigma$ of 1.6. Using a Mie Code (Mie scattering routines, 2018), the aerosol extinction coefficients
at three wavelengths (449, 755 and 1545 nm) were then calculated for this synthetic PSD. Afterwards, the same size retrieval
method as described in Sect 2.2 was used to retrieve $r_{med}$, $\sigma$ and the number density $N_0$. In Fig. 8 a), this retrieved PSD is
plotted with a thinner orange line. It is evident, that the retrieval using the lookup table works very well in such an idealized
case without any measurement uncertainties, when the true and the assumed shape of the PSD match, since both curves are
basically identical.

For Fig. 8 b), a bimodal distribution was assumed as the truth (blue curve), with $r_{med}$ and $\sigma$ values of 90 nm and 1.4 for
the first mode and 350 nm and 1.15 for the second mode. The coarse mode fraction (number density of second mode divided
by number density of first mode) here is chosen as 0.18, i.e. larger than usual. As before, the extinction coefficients at three
wavelengths are calculated and the PSD parameters are retrieved from those, with a monomodal assumption. The resulting
size distribution is again shown as an orange curve. The extinction coefficient, effective radius and number density of both the
bimodal and monomodal distribution is given in the plot, also in blue and orange respectively. The orange curve now stays
close to the second mode instead of the much larger first mode. This is because the extinction efficiency and the extinction
cross section of sulfate aerosols in this size range increase with increasing radius of the particle, which in this case leads the
second mode to dominate the extinction signal produced by the aerosol population. A less extreme example and, based on the
bimodal fit data of the Kiruna OPC measurements, much more common example is shown in Fig. 8 c), where the second mode
is even smaller relative to the first mode, with a coarse mode fraction of roughly 0.03. What is seen here is characteristic of
cases that are in between Fig. 8 a) (monomodal) and Fig. 8 b) (very strong second mode), where the retrieved monomodal PSD
will be influenced by both the first and second mode and its peak will fall somewhere in between both. The larger the second
mode is, the more the retrieval will be dominated by it. With an even smaller second mode than in Fig. 8 c), the retrieval would
be closer to the first mode again and the number density would be closer to the truth.





Figure 8. Retrieved aerosol PSD (orange) from aerosol extinction ratios calculated with a Mie Code from an exemplary, synthetic PSD (blue), simulating the effect of a wrong monomodal assumption. (a): The synthetic, "true" PSD is monomodal, like the retrieved PSD. (b): The "true" PSD is bimodal. (c): Smaller second mode than in (b).

This implies, that the results of the PSD retrievals of this work (and in the end of all remote sensing retrievals of stratospheric aerosol size) have to be interpreted as revealing information about only a certain part of the stratospheric aerosol PSD. This being the part of the size distribution, that mainly contributes to the scattering signal, i.e. very roughly, the larger particles. Still, as discussed in Wrana et al. (2023) this does not call into question the qualitative observation of smaller average stratospheric aerosol size after certain volcanic eruptions, since this observation can only plausibly be explained by an increase in the number



of smaller aerosol particles, albeit an increase in the number of particles that are at least big enough to impact the optical signals measured through remote sensing, if there is no large error in the assumptions on particle composition.

In cases, where the truth comes close to a monomodal lognormal size distribution, this bias is basically non-existent. Sadly, it is at least highly difficult, if not impossible, to measure the size distribution of stratospheric aerosols from remote sensing with sufficiently high size resolution to effectively eliminate this source of error. This is where in situ measurements and model simulations can play an important role to increase our knowledge of the circumstances under which stratospheric aerosols can be described well by a monomodal lognormal size distribution or other PSD shapes.

It is important to keep this possibly big source of errors in mind, when using remote sensing retrieval data of stratospheric aerosol size. Besides impacting other fields of research this issue has strong implications for the feasibility of hypothetical solar radiation management experiments that involve the planned enhancement of the Junge layer (Niemeier et al., 2013). For such experiments it would be critical to be able to monitor the size and, importantly, the surface area of the aerosol particles in the artificially enhanced stratospheric aerosol layer to be able to accurately predict sedimentation rates and therefore the lifetime of the aerosol particles in the stratosphere. As described in this section, the possibility to do this would be severely limited by the missing knowledge of the actual shape of the stratospheric aerosol PSD and therefore of the smaller particles which also compromises the ability to learn about the aerosol surface area.

## 6  Conclusions

For this work, particle size distribution parameters of stratospheric aerosols were retrieved from the SAGE III/M3M solar occultation data set. The evolution from 2002 to 2005 of the most useful parameters was shown and analyzed.

The aerosol size was retrieved using the three-wavelength extinction approach discussed by Wrana et al. (2021), that is viable due to the broad wavelength spectrum of the SAGE III instrument. Because of it no PSD parameter of the retrieved monomodal lognormal aerosol size distribution had to be assumed. In combination with the applicability of said method the SAGE III/M3M stratospheric aerosol data set is special, because it covers a time frame (up to 2004) that is close to background conditions, i.e. not strongly affected by volcanic eruptions and major wildfires. This makes the data useful for validation of other remote sensing stratospheric aerosol size retrieval data sets as well as for microphysical aerosol models that are coupled to climate models.

Also, due to the platform's orbit and the solar occultation geometry the data set does not cover the tropics but instead covers mid to high latitudes, in the northern hemisphere up to $80\,°\text{N}$. This makes it possible to observe some high latitude phenomena, such as polar stratospheric clouds. PSCs were observed in the northern winters of 2002/2003 and 2004/2005, when temperatures were especially low. Starting in the same cold northern winters and lasting for almost a year a reduction in the average size of aerosols was found in the lowermost stratosphere. It was manifested in a decrease in the median radius, absolute distribution width and effective radius and an increase in the particle number density. These effects were stronger in the northern hemisphere. The tropical eruptions of the Ruang, Reventador and Manam volcanoes are likely connected to these size reduction signals, which would put these eruptions in line with the Ambae, Ulawun and La Soufrière eruptions, that



happened during the SAGE III/ISS mission and whose effect on aerosol size has been described in Wrana et al. (2023). All of
these volcanic eruptions share similar characteristics, such as a relatively low amount of $SO_2$ emitted (below 1 Tg), a tropical
location and relatively low injection heights, i.e. low temperature of the ambient air.

However, there are other factors that may have contributed to a decreased average stratospheric aerosol size. Namely the
springtime condensation nuclei (CN) layer, which is forming seasonally in the polar winter stratosphere and can increase the
number of small particles. This may also be linked to meteoric smoke particles (MSPs), which form in the mesosphere, are
transported towards the winter pole and sink to lower altitudes in the stratosphere while interacting with the sulfate aerosols
of the Junge layer and thereby growing larger. This could lead to them growing to measurable size and to an overall smaller
retrieved size of the stratospheric aerosol PSD. For the aerosol size reduction found in 2003 and 2005 this is only relevant for
the northern hemispheric measurements, since the southern hemispheric measurements happen at lower latitudes.

In addition, balloon-borne in situ data from Kiruna, Sweden, was compared to collocated SAGE III/M3M aerosol size
retrieval data in order to validate the latter. Above roughly 15 km altitude, there is a good to very good agreement between both
data sets in effective radius, absolute mode width and number density. Below this altitude level, there seems to be a systematic
difference, with the SAGE III/M3M data set retrieving larger particle sizes and lower number densities than the in situ data.
However, the number of suitable collocations was very limited due to the sparse sampling of both data sets, therefore only
allowing for a limited comparison and validation.

Finally, the effects of the necessary assumption of a certain shape of the retrieved size distribution on the retrieval data were
discussed and illustrated. This is not only an issue for the data set presented in this work, but for aerosol size retrievals from
remote sensing instruments in general, since one way or another they all have to deal with a limited amount of information in
their measurements forcing a simplification of the retrieval, usually by applying some sort of analytical function to the PSD
retrieval. This stands in contrast to the reality of in detail highly variable aerosol PSDs. Since in the typical size range of
stratospheric aerosol larger particles tend to have a higher scattering cross section, even seemingly small deviations from the
assumed size distribution shape can have a strong effect on, e.g., the peak position or the area under the retrieval curve. In the
case of the assumed monomodal lognormal distribution in this work this means, for example, that the existence of a second
peak in the true PSD can lead to a strong underestimation of the number density in the retrieval. In light of this, the important
role of in situ measurements should be stressed, since through them it is possible to retrieve the size distribution of stratospheric
aerosols with a much higher particle size resolution than with remote sensing instruments.

*Data availability.* The data published in this manuscript can be obtained upon request to the first author. The SAGE II and SAGE III/M3M
data was obtained from the NASA Earthdata Atmospheric Science Data Center (https://eosweb.larc.nasa.gov). The in situ data are available
at https://doi.org/10.15786/c.6379371.v1.



*Author contributions.* FW handled the different data sets, carried out the particle size retrieval and created the figures. FW and CvS discussed and interpreted the results and proofread the manuscript. TD provided the OPC data. CL helped to improve the retrieved aerosol size data set. LW helped with the understanding of the intricacies of the SAGE III/M3M data set and the interpretation of retrieval data signals. All authors contributed to the writing of the manuscript.

*Competing interests.* The authors declare that they have no conflict of interest.

*Acknowledgements.* This work was funded by the Deutsche Forschungsgemeinschaft (project VolARC of the DFG research unit VolImpact FOR 2820, grant no. 398006378). We also acknowledge support by the University of Greifswald and thank the Earth Observation Data Group at the University of Oxford for providing the IDL Mie routines used in this study. The Kiruna in situ aerosol measurements were supported by grants from the US National Science Foundation and National Aeronautics and Space Administration.



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
