# Peer review of "Variability of stratospheric aerosol size distribution parameters between 2002 and 2005 from measurements with SAGE III/M3M"

_EGUsphere, 2024_

## Referee Comment (RC2)

Review of Wrana et al. manuscript submitted for publication in ACP, "Variability of stratospheric aerosol
size distribution parameters between 2002 and 2005 from measurements with SAGE III/M3M"

This manuscript presents an analysis of large-scale variations in particle size distribution (PSD) within
the stratospheric aerosol layer, as derived from satellite measurements of aerosol extinction across
multiple wavelengths.

The stratospheric aerosol PSD variations, from 4 years of the SAGE-III/M3M record (2002-2005), span the mid-latitudes
and high-latitudes of both hemispheres, and with the satellite's orbit measuring at high-latitudes through
the polar winter season, represent a substantial and ground-breaking analysis.

The analysis includes three modest-SO2 large-magnitude explosive tropical eruptions, finding these smaller eruptions
(small compared to Pinatubo and Hunga) cause an increase in particle number, but a reduction in particle size,
The high-latitude analysis also identifies a clear decrease in particle size during polar winter, likely associated with
either the influx of meteoric smoke particles (e.g. Curtius et al., 2005; Weigel et al., 2014) and/or new sulphate aerosol
particle formation in late-winter/early-spring (e.g. Campbell and Deshler, 2014).

The manuscript is mostly very well-written, particularly the results section 3, with excellent discussion of the potential
drivers for the observed PSD variations, and the topic will be of substantial interest to the stratospheric aerosol
community.

There are a few paragraphs in earlier parts of the manuscript (particularly section 2.2) that require some minor changes,
and some terminology requires sharpening up slightly -- e.g. text referring to "the solar occultation data set" needs
to be re-worded to be specific to the aerosol extinction retrievals, and the text "aerosol size retrieval" is not really
appropriate, since the aerosol extinction is already a retrieval, and the wording needs to be clear the PSD paramaeters
are derived from the aerosol extintion retrieval -- i.e. "derived aerosol size product" or "derived PSD parameters".

However, this really is an excellent analysis, and once the set of minor revisions below are attended to, this will
represent an important and valuable paper, very suitable for publication in Atmospheric Chemistry and Physics.

Main Minor Revisions
* * *
M-MR-1) "The solar occultation data set" -- Abstract line 1, section 1, line 56, section 2.2, line 108 (and elsewhere)

As mentioned above, the text refers to the SAGE-III/M3M data analysed as "the solar occultation data set", but this is
not sufficiently identifying the sub-set of the SAGE-III data being analysed. Remember that SAGE-III measures not only
aerosol but also stratospheric ozone, NO2 and water vapour, and all of these SAGE-III data-products are measured applying
the solar occultation technique.  Please refer to "the multi-wavelength aerosol extinction data set" or
"stratospheric aerosol data set" to be clear it's the aerosol extinction that's being analysed.

For the Abstract line 1, suggest to change "derived from the solar occultation data of the" instead to "derived from
the multiple-wavelength aerosol extinction retrevals of the"

For section 1, line 56, change "is investigated using the solar occultation data set of the" instead to
"is investigated  using multiple-wavelength aerosol extinction retrevals of the"

For section 2.2, line 108, change "The SAGE III/M3M solar occultation data set is used in this work to derive parameters..."
instead to "The SAGE III/M3M multiple-wavelength aerosol extinction data set is used in this work to derive parameters...",

M-MR-2) "for the retrieval of all parameters" (Abstract line 3), "the aerosol size retrieval" -- section 2.2 lines 118 &
127, and "the retrieval method" on section 2.2 line 113.   This is the other "main minor-revision",  to avoid the word
"retrieval" when referring to the particle size parameters.

The individual-wavelength aerosol extinction is itself a retrieval, and although aerosol extinction at ~1 micron is quite
"clean" in relation to the majority of the extinction being aerosol, for the other wavelength aerosol extinctions analysed,
there is still some uncertainty even within the aerosol extinction (in relation to other species contributing to the
measured extinction at that wavelength, see e.g. McCormick (1987) Figure 4 and Chu and McCormick (1979) Figure 3).

More generally, when the particle size is derived from multiple aerosol extinctions at different wavelengths, it is then
no longer appropriate to refer to the derived size products as a "retrieval".    Please check through the manuscript, and
change "retrieval" instead to "derived product" or "derived PSD parameters" or similar where referring to the size.a

For Abstract line 3, change "allows for the retrieval of all parameters controlling the assumed monomodal lognormal..."
instead to "enables to derive a best estimate for the 3 parameters within an assumed monomodal lognormal..."

For section 2.2 line 118, change "a common assumption in stratospheric aerosol size retrievals from satellite measurements"
to "a common assumption when deriving aerosol size distribution from satellite measurements".

For section 2.2 line 121, change "there is not enough independent spectral information in satellite measurements to
facilitate a retrieval with a more complex model".  That's roughly correct, but it's more to do with non-uniqueness
of the solution, i.e. the derivation to particle size is argued by some to not be sufficiently well-defined to warrant
a more complex particle size model. However, this is a much debated issue, and for example Thomason et al. (2008)
used a bimodal size distribution assumption within their derivation of Surface Area Density and PSD parameters.
In-situ optical particle measurements clearly show that after volcanic eruptions there is clearly a separate mode
with larger volcanic aerosol (see e.g. Deshler, 2008, Figure 4), and the text here needs to be changed
to reflect that there are differing opinions on this.

This sentence (118 to 121) is already quite long , and then to clarify this, suggest to re-structure into 2 sentences.
Within lines 118-119, please delete "on the one hand", and put full-stop after "true conditions".
And then please add "However, in-situ optical particle measurements after the Pinatubo eruption clearly show the presence
of a second mode with much larger volcanic aerosol (see e.g. Deshler, 2008, Figure 4), with Thomason et al. (2008)
using a bimodal size distribution assumption within their method to derive Surface Area Density and PSD parameters."

And then pls change ", but more importantly out of necessity, since usually there is not enough independent spectral
information in satellite measurements to facilitate a retrieval with a more complex model" to
"However methods remain largely mono-modal, and it remains unclear whether or not the derivation of particle size
from spectral aerosol extinction satellite measurements is precise enough to warrant applying a more complex
multi-modal particle size model".

For section 2.2 line 139, change "The retrieval of median radius and mode width is independent of..."
instead to "The derived values of median radius and mode width are independent of..."

For section 2.2 line 144, change "from the retrieved median radius..." to "from the derived median radius..."

For section 2.2 line 145, change "can be calculated from the retrieval results" to "can be calculated from the derived PSD
parameters".

For section 2.2 line 155, change "noisy data in the retrieved quantities." to "noisy data in the derived PSD parameters."

M-MR-3) Improve the terms in the equations, and symbols used, within section 2.2 lines 145-154

Please use roman (not italics) for abbreviations "med" and "eff", and the natural logarithm & exponential functions (ln and exp),

For these equations 3 and 4, the abbreviations Rmed and Reff need to have the "med" and "eff" as roman text (not italics), to be clear these are not additional variables or indices within a matrix etc. Please change the "med" and "eff" also to roman style (rather than italics) in the references to Rmed and Reff in the text (e.g. line 148).
Please add "(Reff)" after "effective radius" on line 145, to introduce the abbreviation, at that first use.
Please also add "aerosol particle" before "effective radius" (line 145) to be clear this is not cloud effective radius.

M-MR-4) Change text "the size of individual aerosol particles can be measured" to avoid potential confusion re: the OPC instrument.

The Wyoming OPC is described to a good level of detail in Deshler et al. (2003). It does not measure the size of individual particles, but counts the number of particles crossing a beam of white light, via individual pulse-occurrences of forward-scattered light at the 40 degree forward-scattering angle.    The information on lines 179-192 suggest this is understood, but the information on line 174 could lead some readers to misunderstand the operation of the instrument.

Please change "Using an optical particle counter (OPC), the size of individual aerosol particles can be measured. This is a major advantage over satellite measurements, whether the measured signal originates from many different aerosol sizes".

Suggest to replace "Using an optical particle counter (OPC), the size of individual aerosol particles can be measured. This is a major advantage over satellite measurements, whether the measured signal originates from many different aerosol sizes".
instead with "An optical particle counter (OPC) measures the number of aerosol particles at light-scattering sizes, and such observations have provided the ground-truth for stratospheric aerosol satellite measurements since the advent of the SAM-II and SAGE instruments (e.g. McCormick et al., 1977)."

That 2nd sentence could also lead to misunderstanding from some readers, and in fact the OPC is measuring a range of particle
sizes, and whilst with this split-sentence, the statement there is not obviously incorrect, it would be misleading as currently worded.

Suggest to replace "This is a major advantage over satellite measurements, whether the measured signal originates from many different aerosol sizes", instead to "The number of aerosol particles is a key microphysical property (e.g. for particle growth),
and although PSD parameters can be derived from satellite, comparisons to in-situ OPC measurements are required to validate and refine aspects of the techniques (e.g. Oberbeck et al., 1989; Hervig and Deshler, 2002)."

M-MR-5) Figure 7 --- please add pale-blue or very-pale-grey shading across the temperature range T = 195K and lower.
This then will have sharing between 21 and 25km in the December 4th 2002 Figures -- add this to all 4 rows of
that central column.   That will then make this much clearer to the reader that this is a PSC being measured there.
(rather than an aerosol enhancement).

Other Minor Revisions
* * *
O-MR-1) Title -- The word "Varibility" somehow to me suggests an analysis across a relatively short period.
Since this is a global-scale analysis of the particle size variations, suggest to replace "Variability of"
instead with "Large-scale variations in the", and delete "parameters" (don't need to specify that in the title).

A change in title is obviously a decision for the authors, but I think "variation" is more scientific than simply "variability", and think that "large-scale" or "hemisphere-scale" should be stated in the title
(to give an indication of the type of variation analysed),   Could possibly add "spatio-temporal" also.

O-MR-2) Abstract lines 1-2 -- change "is shown" and insert "analysed for" before "their evolution".
The Abstract needs to present the MS to be an analysis of the measurements, which it does, very well.

O-MR-3) Abstract line 4 -- change "there were three smaller tropical eruptions" -- whilst it would be OK
to say "smaller eruptions" if caveated to "(than Pinatubo)", in order to reach the stratosphere,
the volcanic eruptions must already be "large-magnitude explosive", so re-word this here.
I think by "small" you mean the amount of SO2 emitted was low/modest, compared to Pinatubo.

Suggest to re-word to "The 2002-2005 stratospheric aerosol layer was mostly at close to background
conditions, but included three moderate-magnitude tropical eruptions (Ruang, Reventador and Manam)".

Then re-word the follow-on sentence instead to refer to the latitudes measured
"The SAGE III/M3M satellite measured only in mid- and high latitudes, but derived PSD parameters
indicate a reduction in particle size after all 3 eruptions (within an increased particle number)".
Or similar wording, to be consistent with the way you choose to summarise this initial finding here.

O-MR-4) Abstract line 6 -- change "Apart from the likely effect of" to "In addition to this effect of..."
(The "Apart from" somehow seems negative, and "In addition" makes clear this is a separate effect.)

O-MR-5) Introduction line 14 -- suggest to add cites to Solomon et al. (2011) and Kremser et al. (2016)
here re: the stratospheric aerosol layer (after "layer of aerosol particles").

O-MR-6) Introduction line 15 -- change "It was first measured" to be clear you mean measured in-situ,
and also give the actual year the first stratospheric aerosol balloon measurements were made, 1957
(see Figure 14 of Junge et al., 1961). Also, it was the high-profile Science paper by Junge et al. (1961)
that brought the recognition of the stratospheric aerosol layer to the general science audience,
and then suggest to cite 1961a and 1961b, as listed in the References below.

Suggest to replace "It was first measured and described by Junge et al. (1961) through balloon measurements"
with "The stratospheric aerosol layer was first measured from high-altitude balloon soundings in 1957
(Junge et al.,1961a, Junge et al., 1961b) ...

O-MR-7) Section 2.1, line 79 -- There needs to be a cite to a paper for the SAGE III/M3M aerosol
measurements here, and suggest to cite Thomason and Taha (2003) here.

O-MR-8) Section 2.1, line 80 -- Suggest to add mention to SAGE-II here, and then cite the 1987
McCormick paper (which you already cite) in the context of the range of different species measured.

Specifically, suggest to add "essentially the same range of" before "different atmospheric" and
replace "like" with "as SAGE-II (see McCormick, 1987)"

O-MR-9) Section 2.2, lines 109-110 -- Please cite also the original methods that were developed
by Yue et al. (1986) to derive the size/SADparameters from SAGE-II, and the 1996 paper that compared
the original method from Yue86 to the PCA method from Thomason and Poole (1993).

Specifically, suggest to change "is in essence the same as the one used in" with "is similar to the
original methods for deriving size parameters from SAGE-II (see Yue et al., 1986, Yue et al., 1995),
and was used also in..."

O-MR-10) Section 2.2, lines 113-114 -- This sentence is too colloquial (in some places), and suggest to
change "A number of assumptions underly the retrieval method and therefore also have to be kept in mind.."
with "There are a number of assumptions within the PSD parameters derived from SAGE-II, which must be
considered when interpreting the results.".  (delete "later on")

O-MR-11) Section 2.2, line 117 -- Please change "Here, the three parameters controlling the monomodal lognormal
size distribution are the median radius...""  with "In this form, there are 3 parameters that together describe
the aerosol particle size distribution: the median radius..."

O-MR-12) Section 2.2, lines 151-152 -- Please change "It is much more useful than the mode width".
That's a rather subjective statement, and remember they're actually the same quantity, just that one is
an absolute measure of the size variation within the mode, and the other is a relative measure.

I have seen that in previous papers the terminology is established to be "absolute mode width" when
referring to the (absolute) standard deviation.  But the term sigma here would usually be expected
to denote that absolute quantity, the absolute standard deviation.  To avoid confusion, within this paper,
I am requesting to add sub-script g in all instances of the sigma, which here is the geometric standard deviation.

Please change all instances of sigma to be sigma-subscript-g, then being clear when you say "the mode width"
(without "absolute") you actually mean the geometric standard deviation (i.e. a mode width of 1.0 means there
is no variation in size within the lognormal, the mode is mono-disperse).

O-MR-13) Section 2.2, lines 152-153 -- Further to the above, please change "It is more useful than...",
to "The absolute mode width provides the variation in nanometers, with variations then easier to interpret than..."

O-MR-14) Section 2.2, line 155 -- add "for the derived PSD parameters" after "an accuracy parameter" and
change "to exclude noisy data" instead to "to be able to exlcude less reliable data".

O-MR-15) Section 2.2, line 167 -- replace "in 0.5km steps" with "at 0.5km vertical resolution".

O-MR-16) Section 2.2, line 180 -- the reference to "12 size classes" here is mis-leading.
 The 12 size channels are counting particles larger than 12 different minimum sizes
(see Deshler et al., 2019). So please change "in 12 size classes" to "larger than 12 size-cuts
(see Deshler et al., 2019)".

O-MR-17) Section 2.2, line 181 -- replace "For this, the scattering of white light by, ideally, single aerosol particles
is measured at an angle of 40o relative to the incident light of the incandescent lamp." with
"The instrument measured individual pulses of forward-scattered white light at an angle of 40 degrees,
a photomultiplier counting individual aerosol particles."

O-MR-18) Section 2.2, lines 184-185 -- change "Aerosol size is retrieved from the measurements" -- it isn't.
There's a threshold size associated with each "size channel", but that's the lower-limit size for
particles being measured within the number concentration for that size-channel.

I think it's best to delete the entire sentence there beginning "Aerosol size is retrieved..."

O-MR-19) Section 2.2, lines 186-187 -- the wording here re: the CN measurement needs improvement.
A suggested re-wording is to change "Also included are separate measurements of the total..."
with "All Kiruna W-OPC soundings analysed here also included a separate measurement of the total"
(I'm assuming that's the case, please check this, or clarify how many did included CNC as well as OPC).

O-MR-20) Section 2.2, line 187-188 -- correct "glycole" to "glycol" and re-word "forcing the aerosol
particles to grow to a detectable size through the condensation of..." with "aerosol particles larger
than 10nm grown to light-scattering sizes via condensation of..."

O-MR-21) Section 3.3, line 344 -- add "ice" before "frost point" and add also "(~3K below the NAT
frost point)".  That then makes clear, that the STS formation temperature is interim between
TICE and TNAT.

O-MR-22) Section 3.3, line 348 -- suggest to replace the Tritscher et al. (2021) review article cite
with the studies of Hoyle et al. (2013) and Engel et al. (2013).

O-MR-23) Section 3.3, line 356 -- please change "plot" to "in Figure 6"

O-MR-24) Section 3.3, line 357 -- please change "of perturbed aerosol extinction"
instead to "of strongly elevated aerosol extinction".

O-MR-25) Section 4, line 377 -- please insert "aerosol" before "extinction coefficient".

O-MR-26) Section 4, line 379 and Figure 7 -- please add labels "a)", "b)" etc. to the sub-panels
in Figure 7m and change "topmost" to refer to "Figure 7a" etc.

O-MR-27) Section 4, line 385 -- change all instances of "collocations" to "co-locations"

References
* * *
Campbell and Deshler (2014)
Condensation nuclei measurements in the midlatitude (1982,Äì2012) and Antarctic (1986,Äì2010) stratosphere between 20 and
35km
J. Geophys. Res. Atmos, vol. 119, 137,Äì152, https://doi.org/10.1002/2013JD019710.

Chu and McCormick (1979)
Inversion of stratospheric aerosol and gaseous constituents from spacecraft solar extinction data in the 0.38-1.0 micron
wavelength region, Applied Optics, vol. 18, no. 9, 1404-1413, https://doi.org/10.1364/AO.18.001404

Curtius et al. (2005)
Observations of meteoric material and implications for aerosol nucleation in the winter Arctic lower stratosphere derived
from in-situ particle measurements,
Atmos. Chem. Phys., 5, 3053‚Äì3069, https://doi.org/10.5194/acp-5-3053-2005

Deshler (2008)
A review of global stratospheric aerosol: Measurements, importance, life cycle, and local stratospheric aerosol
Atmospheric Research, vol. 90, 223‚Äì232, https://doi.org/10.1016/j.atmosres.2008.03.016

Deshler et al. (2019)
Retrieval of Aerosol Size Distributions From In Situ Particle Counter Measurements: Instrument
Counting Efficiency and Comparisons With Satellite Measurements
J. Geophys. Res.: Atmos, 124. https://doi.org/10.1029/2018JD029558

Engel et al. (2013)
Heterogeneous formation of polar stratospheric clouds ‚Äì Part 2: Nucleation of ice on synoptic scales
Atmos. Chem. Phys., 13, 10769‚Äì10785, https://doi.org/10.5194/acp-13-10769-2013

Hervig and Deshler (2002)
Evaluation of aerosol measurements from SAGE II, HALOE, and balloon-borne optical particle counters
J. Geophys. Res., vol. 107, no. D3, 4031, https://doi.org/10.1029/2001JD000703

Hoyle et al. (2013)
Heterogeneous formation of polar stratospheric clouds ‚Äì Part 1: Nucleation of nitric acid trihydrate (NAT)
Atmos. Chem. Phys., 13, 9577‚Äì9595, https://doi.org/10.5194/acp-13-9577-2013

Junge, C. E., Chagnon, C. W., and Manson, J. E. (1961a)
Stratospheric aerosols
J. Meteorol., vol. 18, 81-108, https://doi.org/10.1175/1520-0469(1961)018<0081:SA>2.0.CO;2

Junge, C. E., Chagnon, C. W., and Manson, J. E. (1961b)
A worldwide stratospheric aerosol layer, Science, 133, 1478‚Äì1479, https://doi.org/10.1126/science.133.3463.1478-a.

Kremser et al. (2016)
Stratospheric aerosol‚ÄîObservations, processes, and impact on climate
Rev. Geophys., 54, https://doi.org/10.1002/2015RG000511

McCormick et al. (1979)
Satellite studies of the stratospheric aerosol
Bulletin of the American Meteorological Society, vol. 60, no. 9, pp. 1038-1046
https://doi.org/10.1175/1520-0477(1979)060<1038:SSOTSA>2.0.CO;2

McCormick (1987)
SAGE-II: An overview
Adv. Space Res., vol. 7, no. 3, 219-226, https://doi.org/10.1016/0273-1177(87)90151-7

Oberbeck et al. (1989)
SAGE II Aerosol Validation: Selected Altitude Measurements, Including Particle Micromeasurements
J. Geophys. Res., vol. 94, no. D6, 8367-8380,  https://doi.org/10.1029/JD094iD06p08367

Solomon et al. (2011)
The Persistently Variable ‚ÄúBackground‚Äù Stratospheric Aerosol Layer and Global Climate Change
Science, vol. 333, 866-870, https://doi.org/10.1126/science.1206027

Thomason and Taha (2003)
SAGE-III aerosol extinction measurements: Initial results
Geophys. Res. Lett., vol. 30, no. 12, 1631, https://doi.org/10.1029/2003GL017317.

Thomason and Poole (1993)
Use of Stratospheric Aerosol Properties as Diagnostics of Antarctic Vortex Processes
J. Geophys. Res,, vol. 98, no. D12, 23,003-23,012,
https://doi.org/10.1029/93JD02461

Weigel et al. (2014)
Enhancements of the refractory submicron aerosol fraction in the Arctic polar vortex: feature or exception?
Atmos. Chem. Phys., 14, 12319‚Äì12342, https://doi.org/10.5194/acp-14-12319-2014

Yue et al. (1986)
Retrieval of composition and size distribution of stratospheric aerosols
with the SAGE-II experiment,
Journal of Atmosphere and Ocean Technology, vol. 3, page 371-380
https://doi.org/10.1175/1520-0426(1986)003<0371:ROCASD>2.0.CO;2

Yue et al. (1995)
Aerosol surface areas deduced from early 1993 SAGE II data
and comparisons with stratospheric photochemistry, aerosols,
and Dynamics Expedition measurements
Geophys. Res. Lett., vol. 22, no. 21, 2933-2936,
https://doi.org/10.1029/95GL02941

Engel et al. (2013)